# Revisiting the van der Waals Epitaxy in the Case of (Bi_0.4_Sb_0.6_)_2_Te_3_ Thin Films on Dissimilar Substrates

**DOI:** 10.3390/nano12111790

**Published:** 2022-05-24

**Authors:** Liesbeth Mulder, Daan H. Wielens, Yorick A. Birkhölzer, Alexander Brinkman, Omar Concepción

**Affiliations:** MESA+ Institute for Nanotechnology, University of Twente, 7500 AE Enschede, The Netherlands; liesbethmulder.lm@gmail.com (L.M.); d.h.wielens@utwente.nl (D.H.W.); y.a.birkholzer@utwente.nl (Y.A.B.); a.brinkman@utwente.nl (A.B.)

**Keywords:** thin films, topological insulators, molecular beam epitaxy, van der Waals epitaxy, (Bi_1−*x*_Sb*_x_*)_2_Te_3_

## Abstract

Ultrathin films of the ternary topological insulator (Bi${}_{0.4}$Sb${}_{0.6}$)${}_{2}$Te${}_{3}$ are fabricated by molecular beam epitaxy. Although it is generally assumed that the ternary topological insulator tellurides grow by van der Waals epitaxy, our results show that the influence of the substrate is substantial and governs the formation of defects, mosaicity, and twin domains. For this comparative study, InP (111)A, Al${}_{2}$O${}_{3}$ (001), and SrTiO${}_{3}$ (111) substrates were selected. While the films deposited on lattice-matched InP (111)A show van der Waals epitaxial relations, our results point to a quasi-van der Waals epitaxy for the films grown on substrates with a larger lattice mismatch.

## 1. Introduction

Since the theoretical prediction and experimental realisation of binary topological insulators (TIs) such as Bi${}_{2}$Se${}_{3}$ [1,2,3,4], Bi${}_{2}$Te${}_{3}$ [4,5,6], and Sb${}_{2}$Te${}_{3}$ [6], both the electronic and structural properties of this exotic class of materials have been investigated extensively. The properties of the characteristic topological surface states, such as spin-momentum locking and prohibited backscattering, allow TIs to be used as a fundamental building block in the fields of spintronics, quantum computing, and low-power electronics. Additionally, they are also of interest for fundamental research into exotic phenomena such as Majorana fermions [7,8] and axion electrodynamics [9,10]. To achieve those prospects and unlock the intrinsic properties of those special materials, it is of vital importance that the TIs exhibit an excellent crystallinity, smooth surfaces, and a low defect-density.

The energy of the Dirac point of the topological surface state can be tuned within the bulk band gap of the TI, by varying the ratio of Bi and Sb in ternary (Bi${}_{1-x}$Sb${}_{x}$)${}_{2}$Te${}_{3}$ [11]. Bi${}_{2}$Te${}_{3}$ and Sb${}_{2}$Te${}_{3}$ exhibit the same rhombohedral crystal structure, consisting of a layered structure in the [001] direction. Therefore, it is possible to randomly interchange Bi and Sb atoms to form the ternary compound. In this crystal structure, five atomic layers in the sequence -Te-*X*-Te-*X*-Te- (where *X* is either Bi or Sb) form a quintuple layer (QL) with a height of approximately 1 nm. Atoms within the QL are covalently bonded, but adjacent QLs are bonded via weaker van der Waals (vdW) forces. It is generally stated that these relatively weak interatomic interactions allow the film to be grown by vdW epitaxy on a great variety of substrates, even if these exhibit a large lattice mismatch with respect to the film [12,13].

Thus far, successful deposition of these binary and ternary TIs has been reported on various substrates, such as: Al${}_{2}$O${}_{3}$ (001) [11,14,15], GaAs (111) [16,17], InP (111)A [18], Si (111) [19], SrTiO${}_{3}$ (111) [20,21], and Nb-doped SrTiO${}_{3}$ (111) [22]. Reasons to deviate from choosing a lattice-matched substrate could be to allow for electric field gate tuning through the dielectric substrate, or to be able to illuminate TIs from the backside. For the former reason, SrTiO${}_{3}$ (111) is an ideal candidate [20,21] due to its high dielectric constant at low temperatures, and for the latter reason Al${}_{2}$O${}_{3}$ (001) substrates could be an attractive alternative, due to their wide band gap and consequent high transparency over a wide wavelength regime. These oxide substrates are widely used for their well-established chemical, thermal, and mechanical stabilities. In addition, Al${}_{2}$O${}_{3}$ (001) is relatively cheap, and therefore well-suited for future practical wafer-scale applications. An even more exotic use of substrate material is to induce interfacial superconductivity, e.g., by integration of Bi${}_{2}$Te${}_{3}$ on a layer of FeTe [23] or on a cleaved FeTe${}_{1-x}$Se${}_{x}$ bulk single crystal [24]. However, the resulting films usually show both interfacial and bulk defects and the use of alternative methods such as two-step growth [13,25] or the use of lattice-matched buffer layers [26,27,28] has been explored to improve the film quality. All these cases indicate that the films are not strictly growing by vdW epitaxy, and hence it is important to understand up to what degree the TI material will allow for vdW epitaxy on these substrates.

In terms of thin film deposition techniques, molecular beam epitaxy (MBE) has proven to be well suited for the growth of binary and ternary TIs. TI films grown by MBE are generally of high crystalline quality, however, defects such as mosaicity, twin domains, antiphase domain boundaries, and misoriented crystallites are not exceptional [12,29]. No clear correlation between the existence of these defects and the characteristics of vdW growth has been reported.

This paper presents a comparative study of the influence of the substrate on the structural and morphological properties of (Bi${}_{0.4}$Sb${}_{0.6}$)${}_{2}$Te${}_{3}$ (BST) films grown by MBE. To investigate whether (Bi${}_{1-x}$Sb${}_{x}$)${}_{2}$Te${}_{3}$ solely grows by vdW epitaxy, various experiments were carried out to verify and refute the distinct characteristics of this growth mechanism, i.e., the high crystalline quality, a strain-free growth, the absence of dislocations, a rotational alignment with the substrate, and an abrupt and defect-free substrate/film interface, despite a possible lattice mismatch [30].

## 2. Materials and Methods

BST films were grown in an ultrahigh vacuum Octoplus 300 MBE System from Dr. Eberl MBE Komponenten, with a base pressure of about $5.0\times 10^{-11}$ mbar. High-purity bismuth (6N), antimony (6N), and tellurium (6N) are evaporated from standard Knudsen effusion cells and their fluxes are calibrated by a quartz crystal monitor. The growth rate is determined to be 0.07 QL/min with a flux ratio of (Bi + Sb):Te = 1:10, in order to provide a Te-rich environment during growth to ensure a minimal Te vacancy density. The substrate temperature is kept at 225 ${}^{\circ }$C during growth. The individual substrate preparation procedures are described in Appendix A.

In situ X-ray photoelectron spectroscopy (XPS) measurements are conducted using an Omicron nanotechnology surface analysis system, equipped with a monochromatic aluminium source. High-resolution X-ray diffraction (XRD) measurements and reciprocal space maps (RSM) are collected using a Bruker D8 Discover diffractometer. Texture studies are performed by measuring pole figures using a Panalytical X’Pert Pro MRD instrument by mapping the diffracted intensity as the sample is rotated 360${}^{\circ }$ around the azimuthal (φ) axis and tilted from 20${}^{\circ }$ to 80${}^{\circ }$ around the tilt (χ) axis for fixed 2θ Bragg angles. The surface morphology is examined by atomic force microscopy (AFM) using a Bruker Dimension ICON Microscope in tapping mode in air with Bruker TESPA-V2 probes (with a spring constant of 40 N m${}^{-1}$), where the data are analyzed by using the Gwyddion software [31], and by scanning electron microscopy (SEM) using a Zeiss MERLIN instrument. The lamellae preparation and scanning transmission electron microscopy (STEM) measurements have been performed by Hitachi High-Tech (Minato-ku, Tokyo, Japan), JEOL Ltd. (Akishima, Tokyo, Japan) and Thermo Fisher Scientific (Eindhoven, the Netherlands).

## 3. Results and Discussion

### 3.1. Substrate and Material Choice

InP (111)A, Al${}_{2}$O${}_{3}$ (001), and SrTiO${}_{3}$ (111) substrates were selected for this study, which can be passivated or reconstructed to avoid dangling bonds. The in-plane lattice parameters for these substrates are, respectively, 4.14 Å, 4.76 Å and 5.52 Å. Semi-insulating InP (111)A substrates have a relatively rough surface and a rather modest lattice mismatch of 4.3% with respect to (Bi${}_{1-x}$Sb${}_{x}$)${}_{2}$Te${}_{3}$. Here, the lattice mismatch is calculated using
$$\frac{a_{\mathrm{film}}-a_{\mathrm{substrate}}}{a_{\mathrm{substrate}}}\times 100\%.$$

The properties of the rough InP (111)A surface are in stark contrast with the atomically flat terraces, and reconstructed surfaces of Al${}_{2}$O${}_{3}$ (001) and SrTiO${}_{3}$ (111) insulating substrates. The drawback of the latter substrates is, however, that they exhibit a relatively large lattice mismatch of −9.2% and −21.7%, respectively. Nonetheless, crystalline ultrathin (Bi${}_{1-x}$Sb${}_{x}$)${}_{2}$Te${}_{3}$ films were obtained on these three fundamentally different substrates. However, depending on the substrate material, quantitative differences are observed in the type and amount of defects present in the film. The results of our study show that the differences observed with regard to the structural and morphological properties cannot solely be explained by a vdW epitaxial growth mechanism.

Even though Bi${}_{2}$Te${}_{3}$ and Sb${}_{2}$Te${}_{3}$ have the same rhombohedral crystal structure, there are slight variations in the lattice parameters of these materials. Therefore, all (Bi${}_{1-x}$Sb${}_{x}$)${}_{2}$Te${}_{3}$ films were grown with an equivalent Bi:Sb ratio to rule out the influence of the composition-dependent lattice parameter on our comparative study. The samples were transferred under ultrahigh vacuum conditions to an XPS setup to determine the stoichiometry of the films. We verified that all 10 nm (Bi${}_{1-x}$Sb${}_{x}$)${}_{2}$Te${}_{3}$ films exhibit x=0.59±0.04. The collected spectra of the 10 nm BST film grown on SrTiO${}_{3}$ are shown in Appendix A. The spectra for the other samples are equivalent. Since the samples are transferred in situ, all peaks in the survey scan (Appendix A) can be attributed to either Bi, Sb, or Te. The high-resolution elemental scans (Appendix A) were used to determine the stoichiometry. After a Shirley background subtraction, all elements were fitted employing the constraints suited for spin-orbit splitting of the elements. The determined atomic percentages and calculated stoichiometries are presented in Appendix A. For the stoichiometry calculation, we omitted the Te percentage to determine the Bi:Sb ratio.

### 3.2. High Degree of Crystallinity, Consistent with vdW Epitaxy

For all BST films, reflection high-energy electron diffraction (RHEED) patterns were recorded along $\langle 2\bar{1}\bar{1}0\rangle$ at room temperature, after deposition. Figure 1 presents the acquired RHEED patterns of three 10 nm BST film deposited on InP (a), Al${}_{2}$O${}_{3}$ (b), and SrTiO${}_{3}$ (c). The well-defined diffraction features attest to the high crystalline quality of the film. All RHEED patterns show a spacing between the streaks that is equivalent to having an in-plane lattice parameter of 4.3±0.05 Å. The high crystalline quality of the BST is corroborated by performing ex situ XRD measurements, see Figure 1d. The high intensity peaks correspond to the substrates, i.e., 2θ= 26.30${}^{\circ }$ for InP, 41.68${}^{\circ }$ for Al${}_{2}$O${}_{3}$ and 39.96${}^{\circ }$ for SrTiO${}_{3}$. The remaining diffraction peaks match the parallel planes of the rhombohedral BST structure ([32], PDF nr. 98-018-4248), resulting in an out-of-plane lattice parameter of 30.55±0.1 Å for all the samples, which is the value for the relaxed bulk crystal structure. The presence of only the BST (003n) reflections implies that the *c*-axis of the crystals are properly aligned perpendicular to the substrate surface. The presence of Laue fringes is a clear indication of the high crystalline quality of the BST films and confirms the homogeneity of the thickness [33], and the formation of a clean interface between the substrate and the film [29]. The film thicknesses were verified using X-ray reflectivity (XRR) (see Appendix A).

To investigate the potential in-plane strain induced in the BST films by the different substrates, RSMs were obtained in coplanar geometry in grazing-exit configuration. All RSMs, presented in Figure 2, show a projection of the same region in reciprocal space. This region captures both the (0120) and (1019) reflections of BST and the nearest reflections of the substrates, which are the (133), (018), and (112) reflections for InP, Al${}_{2}$O${}_{3}$, and SrTiO${}_{3}$, respectively. The differences observed in the relative film peak intensities can be explained by considering the rotational alignment of the BST film with the substrates, as will be discussed later on. All film peaks in Figure 2 show an undistorted elliptical shape at an in-plane momentum coordinate corresponding to the inverse projected in-plane lattice parameter of the relaxed phase, independent of the substrate. If (a part of) the film was commensurately strained to the substrate in a cube-on-cube fashion, there would be some intensity at the same in-plane momentum coordinate as the substrate, which is not the case for all three substrates studied here. This is also consistent with previous reports [34] where by selecting substrates without dangling bonds like in this work, also called vdW substrates, the strain at the initial growth phase due to the lattice mismatch is suppressed. Therefore, our results rule out the presence of strain in the film and is consistent with the vdW epitaxial growth mechanism.

### 3.3. Morphological Differences, Indicating Quasi-vdW Epitaxy

The highly crystalline nature of the films, with fully relaxed lattice constants, is consistent with the concept of vdW epitaxy. However, for a pure vdW mechanism, one would expect defect-free films, independent of the choice of substrate. Even though the substrate surfaces differ to a great extent, see Figure 3a–c. In contrast to this expectation, the AFM images of the BST films in Figure 3d–i reveal large differences in the surface morphology. In order to obtain a better understanding of the influence of the substrates on the film growth, the surface morphology of BST films of 5 and 10 nm thickness are compared for different substrates. From left to right, the topography of BST on InP (111)A, Al${}_{2}$O${}_{3}$ (001), and SrTiO${}_{3}$ (111) is presented.

The deposition of 5 nm thick BST films on rough InP substrates results in a smooth and uniform film surface (RMS roughness, determined in Gwyddion after levelling the data based on the QLs, of 0.4 nm), which can be seen in Figure 3d. This is expected to be a direct consequence of the vdW epitaxy and small lattice mismatch between the film and the InP substrate allowing for a layer-by-layer growth mode. The height profiles of the AFM images of the 5 nm film, presented in Appendix A, reveal step edges which appear to be incomplete QLs, but are likely to arise from the InP substrate, where subsequent In atoms are separated by 3.38 Å in the [111] direction. By increasing the film thickness, triangular islands start to appear with their distinctive single QL steps. Similar results have been obtained for the growth of Bi${}_{2}$Se${}_{3}$ on InP (111)B [35] and for Bi${}_{2}$Te${}_{3}$ and Sb${}_{2}$Te${}_{3}$ on Si (111) [36,37,38]. These also exhibit a rough surface and diatomic steps of which the height is comparable to those on the InP (111)A substrate.

The AFM image of the 5 nm film on Al${}_{2}$O${}_{3}$, presented in Figure 3e, reveals a relatively rough film surface (RMS roughness of 1.8 nm). Additionally, it shows three-fold symmetric features that are archetypical for films exhibiting these Bi-based rhombohedral crystal structure. The substrate surface is not completely covered by the BST film, as can be seen in the phase diagrams of the atomic force microscopy images in Appendix A. We refer to these uncovered areas as voids and observe that some of these voids act as nucleation sites for randomly oriented crystallites. For thicker films, the triangular islands start to merge, forming a smooth, dense film, but the misoriented crystallites are still present.

Both the 5 nm and 10 nm film on SrTiO${}_{3}$ show a surface morphology that covers the entire substrate, as can be seen in Figure 3f,i. This observation is in stark contrast to the observed voids in the films on Al${}_{2}$O${}_{3}$. The 5 nm film on SrTiO${}_{3}$ exhibits a RMS roughness of 1.4 nm. We observe individual substrate terrace steps in both the surface morphology of the 5 nm and 10 nm BST film, displaying a step height that corresponds to 2.25 Å for single terminated SrTiO${}_{3}$ (111). A height profile, revealing these substrate terrace steps, is presented in Appendix A.

Since the RHEED and XRD results confirm that the BST film has a crystalline phase, this proves that a layered growth is achieved. The surface morphology shows no clear in-plane orientation for the 5 nm film on SrTiO${}_{3}$, which resembles the lack of in-plane orientation observed in the 5 nm film on InP. In contrast to InP, the absence of the rotational alignment becomes more apparent when increasing the film thickness on SrTiO${}_{3}$. Although there is no sign of voids in these films, similar randomly oriented crystallites are observed as on the films on Al${}_{2}$O${}_{3}$. Since these do not exist on the films on InP, we conclude that these crystallites arise due to strain in the first nucleation layer, as a direct result of the lattice mismatch. This is not expected for a solely vdW epitaxial growth.

The presence of misoriented crystallites has previously been reported for Bi-rhombohedral compounds under different nomenclature: such as blobs, islands, spot structures or randomly inclined crystals [12,14,39,40]. They can even be present in devices that show exotic phenomena such as the quantum anomalous Hall effect [39,41]. Despite the observed relation between the existent lattice mismatch and the presence of these crystals, the origin of these defects is still unclear. Kremer et al. [42] and Richardella et al. [12] performed TEM studies in which they found these misoriented grains to be crystalline. These studies discard the possibility that these disoriented crystals are impurities that they exhibit a different crystal structure, or that they consist of a different stoichiometry. By using Auger electron spectroscopy and energy-dispersive X-ray spectroscopy (EDX), they verified that the crystals had the same composition as the rest of the film. By employing XRD, Ferhat et al. [43] and Mu et al. [40] proved by the presence of polycrystalline BST diffraction peaks that the crystallites are, in fact, the same material but in an unfavourable orientation. In a polycrystalline BST sample, the 2θ diffraction angle of the (015) plane, is very close to the diffraction angle of the (009) plane. Therefore, by using only the diffractogram in Figure 1d, it is not possible to rule-out the presence of the (015) plane in the BST diffractograms entirely. Even though we have not found any indication of the misoriented crystallites in the symmetric XRD scans, the AFM results for the BST films on Al${}_{2}$O${}_{3}$ and SrTiO${}_{3}$ do correspond with the BST misoriented grains as observed in other studies [12,14]. Additionally, the electron backscatter diffraction (EBSD) analysis, which is performed to visualise the in-plane crystal orientation of the films, also showed some amount of misoriented phases, verifying the deviant crystallite orientation (see Appendix A).

According to the Frank-van der Merwe model [44], layer-by-layer growth can only be achieved in case of atomic diffusion and a film to substrate interaction. By performing the thin film growth at the correct temperature and using a substrate with a low lattice mismatch, it is possible to achieve this growth mechanism, as we report in the case of the films grown on InP. Despite the optimisation of the growth temperature and the intrinsically weak interaction between QLs, the large lattice mismatch of BST with respect to both the Al${}_{2}$O${}_{3}$ and SrTiO${}_{3}$ substrate gives rise to three-dimensional island growth. This is in accordance with the Volmer–Weber model [44]. Our results provide additional evidence that the growth of Bi-rhombohedral compounds cannot be considered to be completely consistent with vdW epitaxy on our substrates with large lattice mismatch. This allows us to define the growth mechanism of our BST films as quasi-vdW epitaxy.

### 3.4. Employing Quasi-vdW Epitaxy to Enhance Single Domain Rotational Alignment with the Substrate

Even in the case of pure vdW epitaxy, there is still an inevitable interaction between the film and the substrate. This interaction induces an in-plane alignment of the crystal structures of the film and the substrate [30]. For BST and materials with a similar crystal structure, this in-plane alignment allows for the presence of the characteristic so-called twin domains [12]. When there is only a small interaction between the film and the substrate, the crystalline arrangement within the first QL has two possible orientations with equal probability. This results in grain boundaries with 60${}^{\circ }$ in-plane rotations of the crystal structure around the *c*-axis, referred to as twin boundaries. These boundaries could potentially lead to an enhanced bulk conduction and are therefore detrimental for the film’s applications [17]. For this reason, a suppression of the presence of one of the twin domains is highly desired. In Figure 3h, the AFM image of the 10 nm film deposited on Al${}_{2}$O${}_{3}$ shows that the triangular features are almost exclusively oriented in the same direction, indicating the presence of a single domain, free of twinning. If the growth mechanism would be described solely by vdW epitaxial growth, it would not be possible to suppress one of the domains. Texture studies are performed by measuring pole figures to verify this hypothesis. These measurements allow for the in-plane orientation analysis of the crystals, i.e., to study the presence of the domains as well as the in-plane alignment with the substrate.

The diffraction spots in Figure 4 correspond to the BST {1010} reflections and confirm the aforementioned *c*-axis orientation of the crystals perpendicular to the substrate surface. The azimuthal angles of the spots coincide with those of the substrate (see Appendix A), which is a clear indication of an epitaxial textured growth and characteristic for vdW epitaxial growth. The suppression of one of the twin domains becomes apparent when only three spots in the pole figure are present, such as shown for the films deposited on Al${}_{2}$O${}_{3}$, see Figure 4b. On the contrary, for the rough InP surface, for which the growth takes place by vdW epitaxy, no twin suppression is expected. This is confirmed by the pole figure revealing much less preference for the nucleation of one of the twin domains over the other, see Figure 4a. Similar results were obtained using EBSD (see Appendix A). However, Guo et al. [45] have shown that a domain suppression in Bi${}_{2}$Se${}_{3}$ epilayers deposited by MBE is possible on InP (111)A substrates as well, by enabling a step-flow growth mode by using vicinal InP (111) substrates. The pole figure from the film deposited on SrTiO${}_{3}$, presented in Figure 4c, shows that the film is built up out of two domains with a substantial mosaic twist. A phi-scan representation of the pole figures can be found in Appendix A, where the ratio between the two domains is clearly visible. From these results, we conclude that the large mismatch of SrTiO${}_{3}$ and BST apparently leads to less ordered, mosaically twisted and multi-domain films.

Note that, in the RSM presented in Figure 2b, an inversion of the (0120) and (1019) reflections of BST is observed. While the (0120) is more intense in the RSM performed on the films deposited on InP and SrTiO${}_{3}$ substrates, the (1019) reflection is more pronounced in the RSM of the BST film deposited on the Al${}_{2}$O${}_{3}$ substrate. According to performed calculations using the WinWulff Stereographic software (see Appendix A) and verified by φ-scans performed on the samples, the reflections of the (0120) plane have the same azimuthal angle as the (133) plane of InP, whereas the (1019) reflections are rotated by 60${}^{\circ }$. Since the film deposited on InP exhibits twin domains, both planes should be visible on the RSM. However, as the intensity of the reflections of the (0120) plane is 26.5 times as high as the (1019) reflections ([32], PDF nr. 98-018-4248), only the (0120) is present in the RSM, see Figure 2a. For the films deposited on both the Al${}_{2}$O${}_{3}$ (001) and SrTiO${}_{3}$ (111) substrates, in case of a single twin domain, only the (1019) plane should appear in the RSMs because the reflections of this plane have the same azimuthal angle as the reflections of the (018) and (112) planes of the Al${}_{2}$O${}_{3}$ and SrTiO${}_{3}$ substrates, respectively. This prediction is verified for the single domain BST film grown on Al${}_{2}$O${}_{3}$, presented in Figure 2b. However, for the film on SrTiO${}_{3}$, both the (1019) and (0120) plane are observed due to the presence of both twin domains, see Figure 2c.

### 3.5. Revealing the Nature of the Film-Substrate Interaction in Quasi-vdW Epitaxy

The final characteristic vdW feature is the lattice mismatch independent sharp substrate/film interface [30]. To assess the local crystallinity at the interface of the film and substrates, we use (high-angle) annular dark field, (HA)ADF, scanning transmission electron microscopy (STEM). The (HA)ADF signal is sensitive to variations in the atomic number Z of the constituent atoms, and therefore provides information about local atomic ordering. However, since our material system does not exhibit a large *Z*-contrast, no elemental information can be deduced from the (HA)ADF STEM images alone. For the purpose of mapping the elemental distribution within the film and near the substrate/film interface, we performed EDX. All presented STEM images were obtained using an acceleration voltage of 200 kV, except for those presented in Figure 5c and Figure 6c, which were acquired using 80 kV in order to limit the specimen damage caused by the highly energetic electron beam.

A lamella cut in the $\left[2\bar{1}\bar{1}0\right]$ or [100] direction can reveal twinning of the crystalline rhombohedral crystal structure. However, for a proper examination of the substrate/film interfaces, we used lamellae cut in the $\left[01\bar{1}0\right]$ or [120] direction, in which the crystal structure exhibits a higher level of order due to its atomic arrangement in the *c*-axis direction.

Figure 5a presents ADF STEM results acquired on a lamella fabricated from a 10 nm BST film on InP, cut in the latter $\left[01\bar{1}0\right]$ direction. We observe the well-ordered QL stacking along the *c*-axis, with vdW gaps separating the subsequent QLs. The interface of the BST film and the InP substrate reveals an ordered single, and sometimes double, atomic layer at the interface, smoothing the relatively rough InP surface. Additional EDX analysis is required to verify the composition of this layer. Borisova et al. [36] previously observed the presence of an interfacial Te layer between Bi${}_{2}$Te${}_{3}$ thin films deposited on Si (111), which is therefore proposed to be a likely scenario for our films as well.

In contrast to the observation we made regarding the defect-free interface between the BST film and the InP substrate, the film interfaces at both the Al${}_{2}$O${}_{3}$, Figure 5b, and SrTiO${}_{3}$ substrate, Figure 5c, reveal what resembles an amorphous interfacial layer with a thickness of about 1 nm. The crystalline, unstrained QL ordering appears directly above this layer, which is in agreement with the results obtained from the RSMs. A similar observation was made by Richardella et al. [12] and Richardson et al. [46] for Cr${}_{x}^{}$-(Bi,Sb)${}_{2-x}$Te${}_{3}$ and V${}_{x}^{}$-(Bi,Sb)${}_{2-x}$Te${}_{3}$ deposited on SrTiO${}_{3}$ by MBE.

The unidirectional diagonal alignment of the atoms in the BST film on Al${}_{2}$O${}_{3}$, see Figure 5b, reveals that one of the twin domains is suppressed. From this, we can conclude that, as for InP, the HAADF results are in agreement with the results of the pole figures and the EBSD analysis. Additionally, this STEM image verifies the presence of screw dislocations within the film, which matches the surface morphology as assessed by AFM, see Figure 3e,h.

We performed elemental mapping on the crystalline region of the BST film on SrTiO${}_{3}$, to investigate the composition of the amorphous interfacial layer. The elemental map acquired using EDX on the cross-sectional lamella of the BST film on SrTiO${}_{3}$ is presented in Figure 6. A HAADF reference scan of the area on which the EDX is performed is presented in Figure 6a.

The elemental maps present the -Te-*X*-Te-*X*-Te- stacking within the QL, but also reveal a striking effect near the substrate interface, see Figure 6b,c. The first QL consists of a higher concentration of Bi than the QL farther removed from the interface, likely due to the Sb diffusion towards the substrate interface.

Within the interlayer, the diffused Sb forms an ordered layer. A similar Sb diffusion was observed on our BST film deposited on Al${}_{2}$O${}_{3}$, presented in Appendix A. Lanius et al. [47] presented a similar observation after fabricating a Sb${}_{2}$Te${}_{3}$/Bi${}_{2}$Te${}_{3}$
*p*-*n* heterostructure. A combined EDX and atom probe tomography study revealed the accumulation of both Sb and Te at the substrate/film interface. This is remarkable since they deposited Bi${}_{2}$Te${}_{3}$ prior to Sb${}_{2}$Te${}_{3}$. Thus, the Sb must have diffused through the whole Bi${}_{2}$Te${}_{3}$ layer to reach the Si (111) surface.

Due to the fact that the effect is observed on our BST films deposited on oxide-rich substrates, we speculate that this effect is caused by an imbalance in the formation energies of Bi${}_{3}$O${}_{4}$ and Sb${}_{2}$O${}_{3}$, where the latter forms easier than the former [48]. Since Si substrates are highly prone to oxidation and present a native oxide surface layer, this imbalance in Bi${}_{3}$O${}_{4}$ and Sb${}_{2}$O${}_{3}$ formation energies might also explain the accumulation of Sb near a Si (111) surface in the study of Lanius et al. [47]. Volykhov et al. [49] present a study concerning the surface reactivity of (Bi${}_{1-x}$Sb${}_{x}$)${}_{2}$Te${}_{3}$ in which they explain the chemistry behind the formation of an amorphous antimony oxide at the film surface. They observe an Sb diffusion from the top two QLs towards the oxidized film surface. Combining the findings of this study with our results on (Bi${}_{0.4}$Sb${}_{0.6}$)${}_{2}$Te${}_{3}$ thin films deposited on oxide substrates allows us to draw the conclusion that the Sb-rich interface is caused by the imbalance in Bi${}_{3}$O${}_{4}$ and Sb${}_{2}$O${}_{3}$ formation energies.

## 4. Conclusions

To conclude, in this research, we have investigated the argument that weak vdW interactions allow for the deposition of high-quality BST film on various substrates by vdW epitaxy, regardless of the lattice mismatch. We present consistent results of various experiments performed to support or contradict this argument.

A comparison of the surface morphology of the BST films deposited on various substrates revealed that (i) a larger lattice mismatch resulted in mosaic twisting of the BST and the additional formation of randomly oriented crystals in the ternary TI films. Furthermore, based on the 2θ−ω diffractograms, we concluded that (ii) the highest structural quality is achieved in BST films deposited on lattice-matched substrates. The RHEED results supported both of these claims. Based on a strain analyses performed by mapping the reciprocal space of the samples, we conclude that (iii) no strain is present within the BST film. This observation is verified by STEM images which did reveal an amorphous interlayer for substrates with a larger lattice mismatch, but, above this layer, an abrupt transition to a well ordered crystal structure. With regard to the rotational alignment, we verified that (iv) epitaxial growth on non-vicinal lattice-matched substrates results in inevitable twinning of the crystal structure. Due to the partial agreement of our results with the definition of vdW epitaxial growth, we argue that BST films grow by quasi-vdW epitaxy on substrates with a large lattice mismatch.

## Figures and Tables

**Figure 1 nanomaterials-12-01790-f001:**
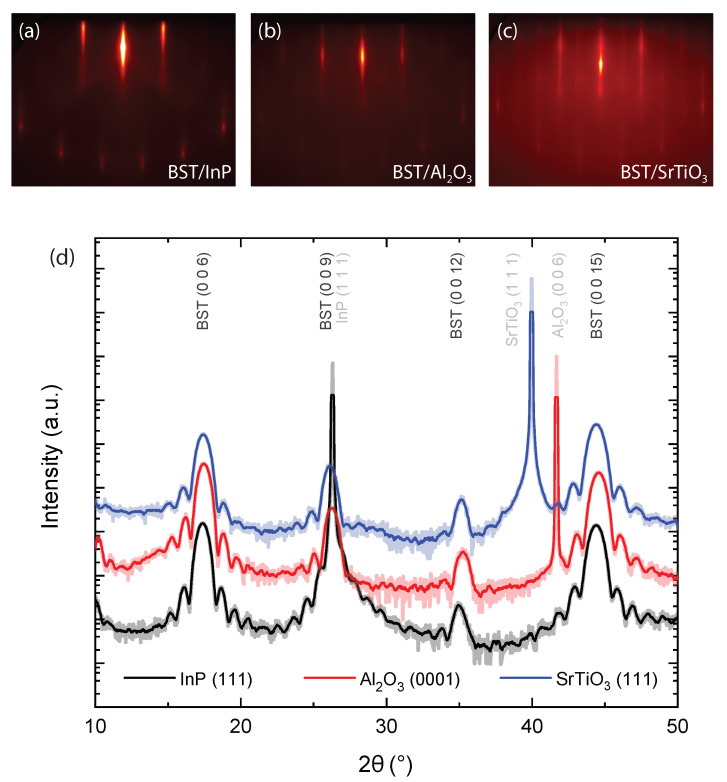
(**a**–**c**) RHEED patterns and (**d**) symmetric 2θ−ω diffractograms of 10 nm BST on InP (111), Al${}_{2}$O${}_{3}$ (001), and SrTiO${}_{3}$ (111) substrates, respectively. The presented RHEED patterns are recorded along $\langle 2\bar{1}\bar{1}0\rangle$ and are representative for other BST films deposited on these substrates. The XRD data have been given an offset for clarity. The solid (semi-transparent) lines present the smoothed (real) data. The Miller indices of the different crystallographic planes (hkℓ) are indicated. The diffractograms and the sharp streaks observed in the RHEED patterns confirm the crystallinity of all three BST films in the out-of-plane direction.

**Figure 2 nanomaterials-12-01790-f002:**
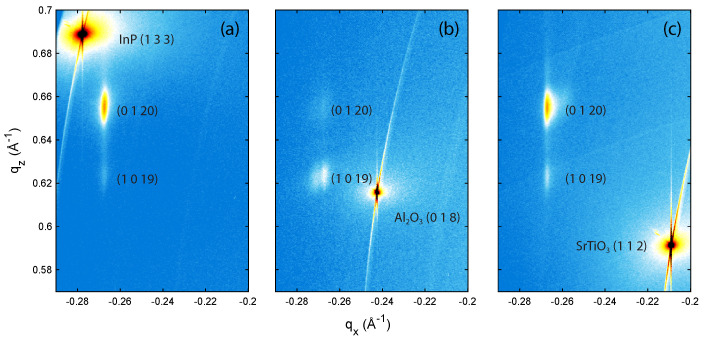
RSMs of 10 nm BST films deposited on (**a**) InP (111)A; (**b**) Al${}_{2}$O${}_{3}$ (001); and (**c**) SrTiO${}_{3}$ (111). All RSMs map out the same region of reciprocal space chosen to cover the three nearest substrate reflections and the reciprocal space position of the BST (0120) and (1019) reflections. The RSMs indicate that there is no significant strain imposed on the film.

**Figure 3 nanomaterials-12-01790-f003:**
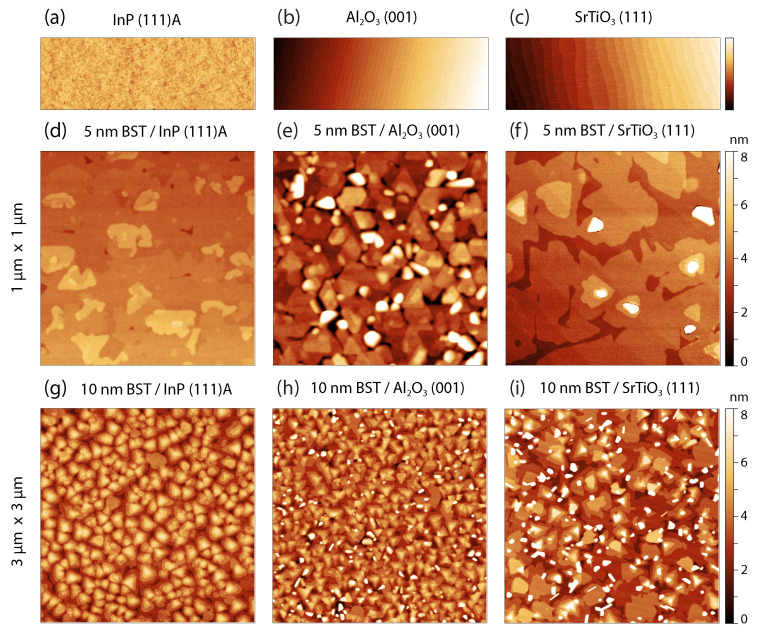
AFM images revealing the surface morphology of the used substrates (**a**–**c**) with a dimension of 1 μm × 3 μm. Surface morphology of 5 and 10 nm BST films deposited on InP (**d**,**g**); Al${}_{2}$O${}_{3}$ (**e**,**h**); and SrTiO${}_{3}$ (**f**,**i**), with a lateral dimension of 1 μm × 1 μm and 3 μm × 3 μm, respectively. The BST films exhibit randomly inclined crystals when deposited on SrTiO${}_{3}$ and Al${}_{2}$O${}_{3}$, whereas it grows relatively smooth on InP. The observed defects are expected to arise from defects in the nucleation layer of the film due to the relatively large lattice mismatch between the film and the SrTiO${}_{3}$ and Al${}_{2}$O${}_{3}$ substrates.

**Figure 4 nanomaterials-12-01790-f004:**
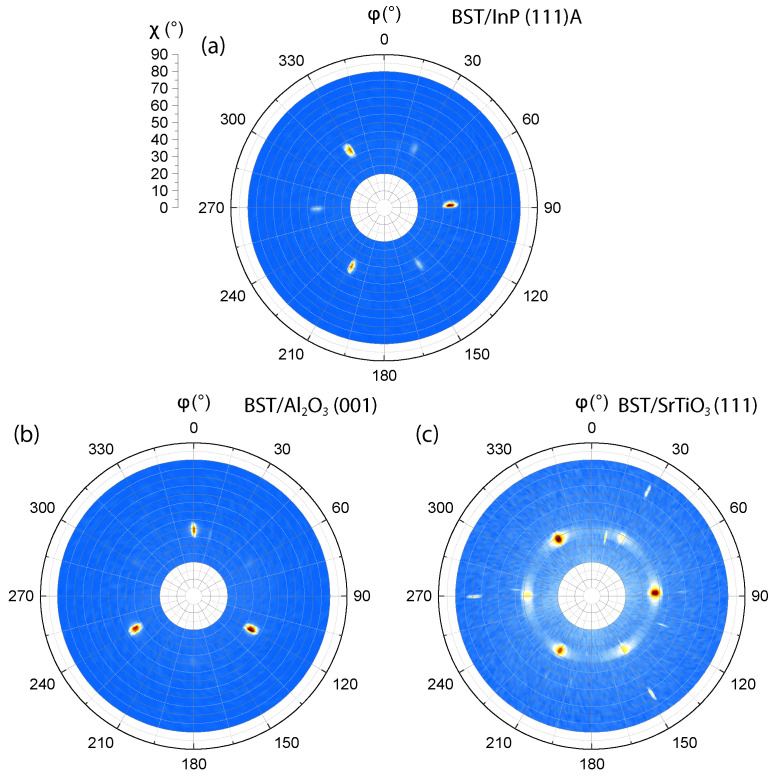
Pole figures taken in symmetric 2θ−ω configuration at $2\theta =38.06{}^{\circ }$ on 10 nm BST films deposited on (**a**) InP (111)A; (**b**) Al${}_{2}$O${}_{3}$ (001); and (**c**) SrTiO${}_{3}$ (111). φ and χ present the azimuthal and tilt angle, respectively. Mapping the BST{1010} of the film on SrTiO${}_{3}$ (**a**) and InP (**c**) clearly shows the existence of the twin domains while a domain suppression is achieved in the film on Al${}_{2}$O${}_{3}$ (**b**).

**Figure 5 nanomaterials-12-01790-f005:**
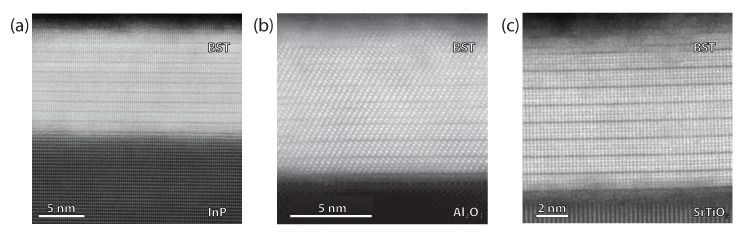
STEM images of the substrate/film interface (**a**) ADF mapping performed on a 10 nm BST film deposited on InP. STEM cross-sectional view along the $\left[2\bar{1}\bar{1}0\right]$ (in-plane) direction, presenting a regular stacking of the QLs and revealing a well-defined substrate/film interface; (**b**) HAADF mapping performed on a 10 nm BST film deposited on Al${}_{2}$O${}_{3}$ presenting a cross-sectional view along the $\left[01\bar{1}0\right]$ (in-plane) direction; (**c**) HAADF STEM cross-sectional view along the $\left[2\bar{1}\bar{1}0\right]$ (in-plane) direction on a 10 nm BST film deposited on SrTiO${}_{3}$.

**Figure 6 nanomaterials-12-01790-f006:**
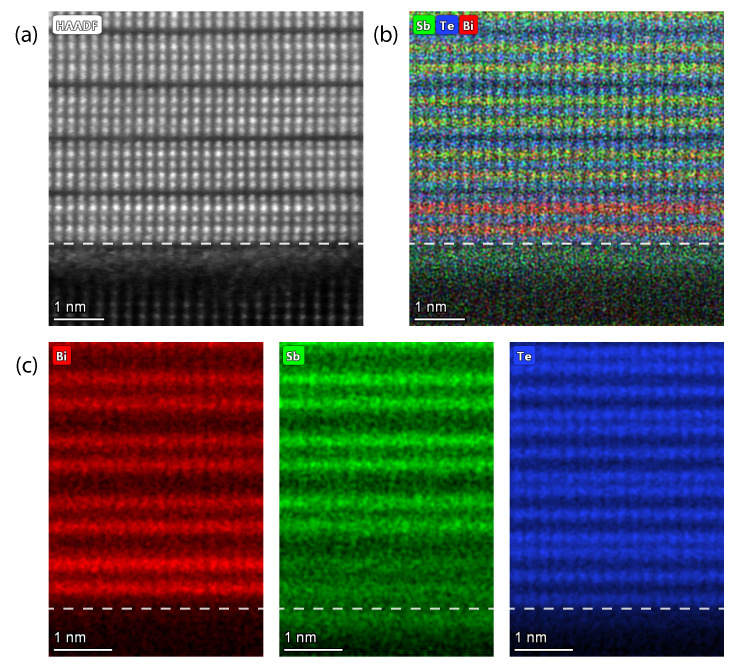
STEM HAADF image (**a**) and EDX maps (**b**,**c**) of lamella presented in Figure 5c, presenting a cross-sectional view along the $\left[2\bar{1}\bar{1}0\right]$ (in-plane) direction on a 10 nm BST film deposited on SrTiO${}_{3}$. The dotted line marks the bottom of the first QL. (**a**) HAADF image of the area on which the EDX mapping is performed; (**b**) overlay of elemental maps; (**c**) individual elemental maps of Bi (red), Sb (green) and Te (blue). The analysis reveals the ordering of Sb in the interfacial layer.

## Data Availability

All data included in this study are available upon request by contact with the corresponding author.

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
