# Peer review of "Revisiting the van der Waals Epitaxy in the Case of (Bi0.4Sb0.6)2Te3 Thin Films on Dissimilar Substrates"

_nanomaterials, 2022, doi:10.3390/nano12111790_

Round 1
Reviewer 1 Report
In this paper, the authors investigated the growth of (Bi1-xSbx)2Te3 (BST) on three different substrates to understand whether the films grow via van der Waals epitaxy. The paper is well written and clearly presented. I have a few comments that might improve the manuscript which are below.
- The authors have neglected some of the literature that helps shed light on the question of whether TI films grow via van der Waals epitaxy. In particular, many groups have shown an improvement of film quality with the use of lattice-matched buffer layers (e.g. 10.1021/acs.nanolett.5b03770; https://doi.org/10.1116/1.5015968; https://doi.org/10.1016/j.jcrysgro.2011.08.029). This would seem to imply that it is already known that these films do not grow strictly by van der Waals epitaxy. Otherwise, a buffer layer would not improve the film quality. Perhaps the authors could add a discussion on this point.
- Likewise, there have been results showing that MBE-grown films are indeed strained for the first few QLs (e.g. 10.1021/acsomega.0c00224; https://doi.org/10.1063/1.4929697). Can the authors comment on why they do not observe this effect in their scans?
- It would be easier to interpret the pole scans if the authors just showed the intensity as a function of angle on a standard plot. The type of plot chosen makes it difficult to see the intensity ratio between the two different domains.
Author Response
Answer to the Reviewers: Revisiting the van der Waals epitaxy in the case of (Bi0.4Sb0.6)2Te3 thin films on dissimilar substrates
First of all, I would like to say on behalf of all the authors, that we are very grateful for the invaluable review of the referees. In the new version, we have tried to consider the different referee's concerns.
Please find a “Revised Manuscript” with the changes made using the “Track Changes” function.
Below, we will address every referee comment:
Reviewer 1:
In this paper, the authors investigated the growth of (Bi1-xSbx)2Te3 (BST) on three different substrates to understand whether the films grow via van der Waals epitaxy. The paper is well written and clearly presented. I have a few comments that might improve the manuscript which are below.
- The authors have neglected some of the literature that helps shed light on the question of whether TI films grow via van der Waals epitaxy. In particular, many groups have shown an improvement of film quality with the use of lattice-matched buffer layers (e.g.10.1021/acs.nanolett.5b03770; https://doi.org/10.1116/ 1.5015968; https://doi.org/10.1016/jcrysgro.2011.08.029). This would seem to imply that it is already known that these films do not grow strictly by van der Waals epitaxy. Otherwise, a buffer layer would not improve the film quality. Perhaps the authors could add a discussion on this point.
There are indeed some indications that the material does not grow strictly by VdW epitaxy. These are also discussed/mentioned in the manuscript. There are reports available that mention alternative methods to improve the crystalline quality, such as the use of a buffer layer or the two-step growth method. However, in this work, we do not intend to focus on the improvement of the crystalline quality of the material, rather the goal was to delve into the concept of vdW epitaxy to clarify the influence of this growth method depending on the substrate used.
To clarify this point, in the main text, line 46, we add:
“However, the resulting films usually show both interfacial and bulk defects and the use of alternative methods such as two-step growth [13,25] or the use of lattice-matched buffer layers [26-28] has been explored to improve the film quality. All these cases indicate that the films are not strictly growing by vdW epitaxy, and hence it is important to understand up to what degree the TI material will allow for van der Waals epitaxy on these substrates.”
- Likewise, there have been results showing that MBE-grown films are indeed strained for the first few QLs (e.g. 10.1021/acsomega.0c00224; https://doi.org/10.1063/1.4929697). Can the authors comment on why they do not observe this effect in their scans?
Our results (RSM/TEM) do not show strain in the first QLs, possibly due to the selection/preparation of the substrates used. We made sure that their surfaces did not have dangling bonds, since these types of substrates are the ones that allow vdW epitaxy. More information about the treatment of our substrates can be found in Section I of the SI. However, for clarity, in line 147 of the main text we add:
“This is also consistent with previous reports [34] where by selecting substrates without dangling bonds like in this work, also called vdW substrates, the strain at the initial growth phase due to the lattice mismatch is suppressed. Therefore, our results rule out the presence of strain in the film and is consistent with the vdW epitaxial growth mechanism.”
- It would be easier to interpret the pole scans if the authors just showed the intensity as a function of angle on a standard plot. The type of plot chosen makes it difficult to see the intensity ratio between the two different domains.
The reviewer is right: the chosen plots do not facilitate the interpretation of the ratio between the domains. However, this type of plot was selected by the authors intentionally. In this work, our aim was not to study the domains or how to suppress them (although it is mentioned). The study focused on the orientation of the planes of the material with respect to the substrate (Figure S6) and between the BST (1019) and (0120) planes (Figure S8). For this purpose, polar graphs are more intuitive.
To visualize the intensity ratio between the two different domains, we added Phi-scans to the SM, see Figure S7, as well as a sentence in line 270:
“A phi-scan representation of the pole figures can be found in the Supplemental Material, where the ratio between the two domains is clearly visible.”

Reviewer 2 Report
The paper reports a study on epitaxial growth of BiSbTe films. The role of different substrates on the growth mechanisms is well discussed along the paper, the comparison of data from different characterization techniques is critically discussed. The conclusions are well supported by data. For all these aspects, I suggest the publication in this form.
Author Response
Answer to the Reviewers: Revisiting the van der Waals epitaxy in the case of (Bi0.4Sb0.6)2Te3 thin films on dissimilar substrates
First of all, I would like to say on behalf of all the authors, that we are very grateful for the invaluable review of the referees. In the new version, we have tried to consider the different referee's concerns.
Please find a “Revised Manuscript” with the changes made using the “Track Changes” function.
Reviewer 3 Report
The manuscript investigates the characteristics of (Bi0.4Sb0.6)2Te3 topological insulators grown by molecular beam epitaxy on different kinds of substrates, i.e. InP (111)A, Al2O3 (001) and SrTiO3 (111). A rigorous investigation has been performed by means of several experimental techniques, leading the authors to propose the van der Waals epitaxial growth for films grown on lattice-matched substrates (InP) and a quasi-van der Waals epitaxial growth for films grown on substrates with higher lattice mismatch (Al2O3 and SrTiO3).
The manuscript is interesting, clear and well written. A couple of minor issues should be addressed as explained below.
1) The X-ray reflectivity (XRR) spectra used to infer the thickness of (Bi0.4Sb0.6)2Te3 films, mentioned in the text, could be reported in the SI.
2) Please correct typing errors at lines 148-150.
Author Response
Answer to the Reviewers: Revisiting the van der Waals epitaxy in the case of (Bi0.4Sb0.6)2Te3 thin films on dissimilar substrates
First of all, I would like to say on behalf of all the authors, that we are very grateful for the invaluable review of the referees. In the new version, we have tried to consider the different referee's concerns.
Please find a “Revised Manuscript” with the changes made using the “Track Changes” function.
Below, we will address every referee comment:
Reviewer 3:
The manuscript investigates the characteristics of (Bi0.4Sb0.6)2Te3 topological insulators grown by molecular beam epitaxy on different kinds of substrates, i.e. InP (111)A, Al2O3 (001) and SrTiO3 (111). A rigorous investigation has been performed by means of several experimental techniques, leading the authors to propose the van der Waals epitaxial growth for films grown on lattice-matched substrates (InP) and a quasi-van der Waals epitaxial growth for films grown on substrates with higher lattice mismatch (Al2O3 and SrTiO3).
The manuscript is interesting, clear and well written. A couple of minor issues should be addressed as explained below.
- The X-ray reflectivity (XRR) spectra used to infer the thickness of (Bi0.4Sb0.6)2Te3 films, mentioned in the text, could be reported in the SI.
The XRR spectra of the BST films were added to the SM file, Figure S2, as well as a note in line 133.
- Please correct typing errors at lines 148-150.
We thank the reviewer for pointing out these typing errors. The errors have been corrected.

Reviewer 4 Report
In their paper "Revisiting the van der Waals epitaxy ...", Liesbeth Mulder and coworkers present a detailed study on growth and epitaxy of a topological insulator on several different substrates with a focus on vdW epitaxy. Diffraction and imaging methods used are described in sufficient detail. The research methodology is sound and the obtained results give detailed insights into the influence of substrate-adsorbate interactions on the level of vdW epitaxy, which might be of use for modeling as well as for possible applications. The paper should therefore be published after some minor revision.
Minor points which need to be addressed:
- line 110: "splitting", not "spitting"
- line 181: "substrate", not "subsrtate"
- Figure 3 and lines 169-175 and 177-183: The authors claim that for Al2O3 the "substrate surface is not completely covered by the BST film." How can they deduce this from the AFM image in Figure 3? Are there any phase contrast images available? Similar for SrTiO3: how can the authors be sure that the entire surface is covered?
- Figure 3: The authors obviously use Gwyddion to process AFM images, but a proper reference is not included.
- Lines 185-193: From the morphology of the grown islands the authors deduce some statements on the rotational alignment. Although this is later substantiated by pole figures, the authors should somehow quantify their statements by, e.g., histograms of crystallite orientations. Why haven't the authors used LEED to characterize the in-plane orientation? Can the authors provide the superlattice matrix for the films? Can they exclude point-on-line or line-on-line epitaxial growth?
- Line 360: Are the results really "self-consistent", and not just "consistent"?
- References: some journal titles are abbreviated, some are not.
- Table S1: What are the error bars for the stoichiometry?
Author Response
Answer to the Reviewers: Revisiting the van der Waals epitaxy in the case of (Bi0.4Sb0.6)2Te3 thin films on dissimilar substrates
First of all, I would like to say on behalf of all the authors, that we are very grateful for the invaluable review of the referees. In the new version, we have tried to consider the different referee's concerns.
Please find a “Revised Manuscript” with the changes made using the “Track Changes” function.
Below, we will address every referee comment:
Reviewer 4:
In their paper "Revisiting the van der Waals epitaxy ...", Liesbeth Mulder and coworkers present a detailed study on growth and epitaxy of a topological insulator on several different substrates with a focus on vdW epitaxy. Diffraction and imaging methods used are described in sufficient detail. The research methodology is sound and the obtained results give detailed insights into the influence of substrate-adsorbate interactions on the level of vdW epitaxy, which might be of use for modeling as well as for possible applications. The paper should therefore be published after some minor revision.
Minor points which need to be addressed:
- line 110: "splitting", not "spitting"
This tying error has been corrected.
- line 181: "substrate", not "subsrtate"
This tying error has been corrected.
- Figure 3 and lines 169-175 and 177-183: The authors claim that for Al2O3 the "substrate surface is not completely covered by the BST film." How can they deduce this from the AFM image in Figure 3? Are there any phase contrast images available? Similar for SrTiO3: how can the authors be sure that the entire surface is covered?
We agree with the reviewer. Therefore, we added two images to the SM (Figure S4), containing the phase information corresponding to the two scans in the main text. These images allow the reader to distinguish between the areas that are covered by the BST and the areas where the substrate is exposed.
- Figure 3: The authors obviously use Gwyddion to process AFM images, but a proper reference is not included.
We agree with this comment and added a note on the use of Gwyddion in line 86 and included the reference.
- Lines 185-193: From the morphology of the grown islands the authors deduce some statements on the rotational alignment. Although this is later substantiated by pole figures, the authors should somehow quantify their statements by, e.g., histograms of crystallite orientations. Why haven't the authors used LEED to characterize the in-plane orientation? Can the authors provide the superlattice matrix for the films? Can they exclude point-on-line or line-on-line epitaxial growth?
To quantify the orientations on the surface, in Figure S5 of the SM we show EBSD results where the orientations of the domains can be observed morphologically.
Unfortunately, we are limited to the use of the LEED technique due to our MBE system does not have it.
Based on the references [https://onlinelibrary.wiley.com/doi/epdf/10.1002/1521-4095%28200102%2913%3A4%3C227%3A%3AAID-ADMA227%3E3.0.CO%3B2-P] and [https://journals.aps.org/prl/abstract/10.1103/PhysRevLett.94.056104] we will like to have the matrix for the films/substrate to identify the specific type of epitaxy but unfortunately, we don’t have this data and definitely, we will consider it in the future.
- Line 360: Are the results really "self-consistent", and not just "consistent"?
The reviewer is correct, and thus we have replaced the former with the latter word.
- References: some journal titles are abbreviated, some are not.
We have corrected this.
- Table S1: What are the error bars for the stoichiometry?
For the XPS setup used in this work, the error has been determined to be in the order of a few percent (see 10.1002/pssr.201800679).
It is important to note that the determination of the exact stoichiometry is not the main goal of our XPS measurements – we rather want to compare the stoichiometry of the BST films deposited on the dissimilar substrates. To minimize the relative error between the measurements, we make sure that data processing of the spectra is performed using a predefined set of constraints (based on the spin-orbit splitting and ratio of peak area). Furthermore, for all samples, the scans were performed using the same settings (same energy resolution, dwell time, no. of repetitions, etc.). A note was added to the SM, Section II.
